# Benzylideneacetophenone Derivative Alleviates Arthritic Symptoms via Modulation of the MAPK Signaling Pathway

**DOI:** 10.3390/molecules25153319

**Published:** 2020-07-22

**Authors:** Bongjun Sur, Mijin Kim, Thea Villa, Seikwan Oh

**Affiliations:** Department of Molecular Medicine, School of Medicine, Ewha Womans University, Seoul 07804, Korea; surzeus@naver.com (B.S.); atthatinstant@naver.com (M.K.); thea.villa13@gmail.com (T.V.)

**Keywords:** arthritis, benzylideneacetophenone, TNF-α, anti-inflammatory, fibroblast-like synoviocytes

## Abstract

The benzylideneacetophenone derivative 3-(4-hydroxy-3-methoxy-phenyl)-1-{3-[1]-phenyl}-propenone (JC3 dimer) was synthesized through the dimerization of JC3. To investigate the inhibitory effects of JC3 dimer, the carrageenan/kaolin (C/K)-induced knee arthritis rat model was used in vivo and rheumatoid arthritis (RA) patient-derived fibroblast-like synoviocytes (FLS) were used *in vitro*. In the C/K rat model, JC3 dimer was given after arthritis induction for 6 days at the concentrations of 1, 5, or 10 mg/kg/day. Manifestation of arthritis was evaluated using knee thickness, weight distribution ratio (WDR), and squeaking test. The levels of prostaglandin E_2_ (PGE_2_), interleukin (IL)-6, and tumor necrosis factor (TNF)-α in the serum of JC3 dimer-treated arthritic rats were also analyzed. Histological examination of the knee joints was also done. For the FLS, the cells were stimulated using IL-1β and concentrations of 1, 5, and 10 μg/mL JC3 dimer were used. The levels of IL-8, IL-6, and PGE_2_ were measured in stimulated FLS treated with JC3 dimer. At days 5 to 6 after arthritis induction, JC3 dimer treatment significantly decreased arthritic symptoms and reduced the inflammation in the knee joints in the histology of knee tissues in C/K-arthritic rats. In stimulated FLS, JC3 dimer suppressed the increase of IL-8, IL-6, and PGE_2_. These findings suggest that JC3 dimer has suppressive effects on arthritis, and that JC3 dimer can be a potential agent for arthritis therapy.

## 1. Introduction

Rheumatoid arthritis (RA) is a chronic inflammatory disease characterized by a persistent inflammatory reaction of the synovial membrane [1]. It is very common among 1–2% of the total population, but the disease cause is still unclear [2]. The chronic inflammatory reaction of the synovial membrane is a major clinical feature that causes cartilage and bone damage in the joints, resulting in disruption of the joints [3]. The pathology of RA is depicted by inflammatory cellular infiltration in the pannus and the joint fluid, giving rise to the destruction of tissue [4]. Not only that, other inflammatory mediators are also involved in RA progression as they regulate the inflammatory response that can be seen in patients afflicted with RA [5]. In addition, the imbalance between pro-inflammatory cytokine and anti-inflammatory cytokine activity can lead to autoimmunity and chronic inflammation, thereby resulting in joint damage [6].

An important cell that can be found in the rheumatoid synovium is the fibroblast-like synoviocyte (FLS). In synovitis, there is the presence of a hyperplastic synovial lining that can be attributed to the RA synovial environment, promoting the dysregulation of FLS apoptosis and the activation of synovial cells. FLS plays an important role in exacerbating arthritis [7]. These cells, especially when activated, are a major source of pro-inflammatory chemicals that can be found in the synovium. Activated FLS secrete inflammatory cytokines and chemokines that contribute to synovium infiltration. FLS also promote joint destruction through the production of matrix metalloproteinases (MMPs) and the formation of pannus, which is an invasive synovial tissue composed of macrophages, FLS, osteoclasts, and lymphocytes [8]. Inflammatory cytokines produced by FLS in the synovium include tumor necrosis factor (TNF) and interleukin (IL)-1. TNF contributes to osteoclast activation and differentiation, and IL-1 can also stimulate synovial cell growth and activation as well as also contribute to osteoclast stimulation [9,10]. Thus, different cellular interactions that are mediated by IL-1 and TNF and the macrophage-produced cytokines are causes that can lead to cartilage damage in RA [11,12]. Other inflammatory mediators involved in RA include IL-6, IL-8, and prostaglandin E_2_ (PGE_2_) [13].

Carrageenan comes from Chondrus spp. and Gigartina spp., which are seaweeds, and it is a sulfated mucopolysaccharide [14]. It has been used in the enhancement of footpad inflammation and paw edema in animal models as well as in progressing the pathological deterioration in an RA animal model [15,16]. Injecting carrageenan intra-articularly into the knee joint results in inflammation, inducing inflammatory mediator production such as TNF-α, cyclooxygenase (COX)-2, and prostaglandin E_2_ (PGE_2_) [17].

Yakuchinone B is commonly used in folk medicine and can be isolated from the seeds of Alpina oxyphylla, which is also a part of the Zingiberaceae ginger family. It is a conjugated 1,4-enone with a phenyl ring, and a part of an important class of natural chalcones that exerts an extensive range of biological activities [18]. Such biological activities include anti-inflammatory [19,20,21], antiviral [22], and antitumor [23] effects. Yakuchinone B was modified in order to synthesize the benzylideneacetophenone JC3 dimer to develop potential anti-inflammatory agents [24]. In this study, by using carrageenan/kaolin (C/K)-induced rat models and FLS from arthritis patients that were stimulated with IL-1β, the effects of JC3 dimer on the inhibition of arthritis were evaluated.

## 2. Results and Discussion

### 2.1. C/K-Induced Arthritis Rats and the Anti-Arthritic Effect of JC3 Dimer

Physical behavioral parameters such as knee morphology, weight distribution ratio (WDR), knee thickness, and squeaking scores were evaluated in order to examine the effect of JC3 dimer on arthritic rats. Histological analysis was also done to observe knee tissue morphological changes and infiltration by inflammatory cells. Knee joint thickness was measured daily as a simple method to evaluate C/K-induced arthritis. The knee thickness of the normal group was about 10 mm until the end of the experiment, but the C/K-induced arthritic group (ART) group showed significant increase throughout the experiment after the injection of C/K. However, in the experimental group treated with JC3 dimer, there was a decreasing trend beginning in the second day after the introduction of C/K, which continued until the last day of the experiment. Particularly, JC3dimer_10 experimental groups showed a significant decrease from day 5 (Figure 1A). To evaluate the degree of pain caused by inflammation after the induction of arthritis, vocalization or squeaking was measured. The squeaking scores while extending and flexing reached a maximum on day 1 post-injection of C/K. The vocalization scores of the normal group were not nearly vocalized until the end of the experiment, but the vocalization scores of the ART group showed maximal vocalization until the end of the experiment after C/K injection. On the other hand, in the ART + JC3dimer_10 (carrageenan/kaolin-induced and 10 mg/kg JC3-treated) group, the vocalization scores from day 5 were significantly lower after C/K injection, but this effect was not observed in the ART + JC3dimer_1 (carrageenan/kaolin-induced and 1 mg/kg JC3 dimer-treated) and the ART + JC3dimer_5 (carrageenan/kaolin-induced and 1 mg/kg JC3 dimer-treated) groups (Figure 1B). Finally, JC3 dimer and its effects on WDR in C/K-induced arthritis rats were examined. The mean WDR at baseline before C/K injection was 50:50. A decrease in WDR was observed a day after inducing C/K arthritis. On day 6, the weight carried by the arthritic legs in the ART group dropped up to 20%. In contrast, in animals that were administered with 5 or 10 mg/kg JC3 dimer, WDR did not reduce as much with the results being especially significant in ART + JC3dimer_10 when compared to the ART group (Figure 1C). In the histological analysis, the NOR group showed no inflammatory signs in the knee, whereas the C/K-induced arthritic group (ART group) showed severe signs of inflammation such as hyperplasia of the synovium, formation of a pannus, infiltration of inflammatory cells, and degradation of cartilage and bone. On the other hand, the group treated with JC3 dimer showed reduced signs of inflammation when compared to the ART group (Figure 1D).

### 2.2. Inflammatory Mediators in the Serum of C/K-Induced Arthritis Rats and the Effect of JC3 Dimer

C/K injection causes an increase of variety of inflammatory cytokines in the synovium, increasing joint destruction and synovial inflammation. In this study, the inflammatory mediators IL-6, TNF-α, and PGE_2_ were analyzed (Figure 2). The C/K-injected group had a significant rise in the serum levels of TNF-α, IL-6, and PGE_2_ production when compared to the NOR group (Figure 2A–C). The JC3 dimer treatment group significantly inhibited the serum levels of the above-mentioned inflammatory mediators dose-dependently. In particular, the decrease in protein levels of TNF-α in the ART + JC3dimer_10 group was close to the normal group (Figure 4A).

### 2.3. Inflammatory Mediator Production in FLS and the Effect of JC3 Dimer

IL-1β (10 ng/mL) was used to stimulate FLS from RA patients. Stimulated FLS had increased mRNA levels of IL-8 and IL-6 and protein levels of PGE_2_, IL-8, and IL-6 in comparison with unstimulated FLS (Figure 3 and Figure 4). JC3 dimer treatment suppressed significantly the increase of mRNA levels of IL-6 and IL-8 as well as its protein levels dose-dependently (Figure 3A,B; Figure 4A,B). In addition, JC3 dimer treatment significantly inhibited the protein levels of PGE_2_ dose-dependently (Figure 4C).

### 2.4. IL-1β-Induced Phosphorylation of p38/ERK/JNK MAPKs in FLS and the Effect of JC3 Dimer

MAPK (Mitogen-activated protein kinase) pathways are important because they mediate signal transduction as a cellular response to extracellular signals in arthritis [25]. To analyze if MAPKs are involved in the inhibition of arthritic inflammation exerted by JC3 dimer, p38, extracellular signal-regulated kinase (ERK), c-Jun N-terminal kinase (JNK) MAPK phosphorylation in IL-1β-treated FLS and treatment with JC3 dimer was analyzed using ELISA (Figure 5). IL-1β-treated cells had a significant increased phosphorylation of p38, ERK, and JNK MAPKs (Figure 5A–C). In the untreated group, significant MAPK phosphorylation was not observed. JC3 dimer treatment (10 μg/mL) significantly inhibited the increased MAPK phosphorylation in stimulated FLS with the greatest significance in p38 phosphorylation, followed by ERK phosphorylation, and then JNK phosphorylation.

In this study, different inflammatory mediators involved in RA were decreased by JC3 dimer both in vivo and in vitro. These results are consistent with other studies involving anti-inflammatory effects in animal models [26]. To demonstrate the anti-arthritic effect of JC3 dimer in vivo, C/K-induced arthritis in SD rats was used. Treatment with JC3 dimer in rats with arthritis showed signs and symptoms of improvement through the measured physical parameters and histological analysis, which indicated alleviation of arthritic manifestation, inflammation, and pain. This is further supported by the decrease of the inflammatory mediators TNF-a, IL-6, and PGE_2_ in the serum of the rats that were treated with JC3 dimer.

IL-1β was used to stimulate FLS in vitro, as it has been used to simulate the proliferation of the cells and growth of the synovium in RA [27]. We found that JC3 dimer significantly decreased the production of pro-inflammatory mediators, IL-6 and PGE_2_, and the angiogenesis mediator, IL-8, in FLS stimulated with IL-1β. Particularly, by inhibiting PGE_2_ production, inflammatory pain and edema were controlled, making it an important mediator in RA [21,28].

As described above, JC3 dimer significantly suppressed the production of IL-8, which is an important angiogenic agent. This result suggests that angiogenesis is potentially inhibited by JC3 dimer. Interestingly, the expression of MMPs, such as MMP-1 and MMP-13, was unaffected by JC3 dimer in IL-1β-stimulated FLS (data not shown). In the pathogenesis of RA, MMPs slow down and restrict the collagen degradation process. Thus, while JC3 dimer has an anti-arthritic effect on inflammation and angiogenesis, it does not affect collagen degradation.

To understand the mechanisms behind these circumstances, FLS intracellular signal transduction was analyzed to investigate the effect of JC3 dimer. MAPKs, through pro-inflammatory cytokines such as IL-1β, can be induced and promote the production and transcription of matrix metalloproteinases, adhesion molecules, and pro-inflammatory genes [29]. It has previously been demonstrated that other than the p38 pathway, the JNK pathway is also required for the production of MMPs in cytokine-activated synovial fibroblasts. The MMPs involved are MMP-1 in patients with RA and MMP-13 in murine inflammatory arthritis [30]. In this study, JC3 dimer specifically inhibited the phosphorylation of p38 and ERK in relation to the intracellular signaling of IL-1β stimulated FLS. As the JNK pathway was the least affected MAPK, it is not surprising the MMP-1 and 13 were also unaffected. Although the exact mechanisms as to why JNK was the least affected MAPK is unknown, this leaves another area for future investigations. Overall, these results suggest that this anti-inflammatory effect of JC3 dimer is mediated through the blockade of p38 and ERK signaling pathways in FLS stimulated with IL-1β.

## 3. Materials and Methods

### 3.1. Compound Synthesis

The synthesis of JC3 dimer was done as previously described [21]. Dimer types of benzylideneacetophenone have been developed in order to produce potential antioxidative agents and anti-excitotoxic compounds, which is in accordance with our program that deals with 1,3-diaryl-2-propen-1 derivative development. With the presence of stoichiometric amounts of c-sulfuric acid in ethanol and from commercially available 4-hydroxy-3-methoxybenzaldehyde and 1,3-diacetylbenzene, 3-(4-hydroxy-3-methoxy-phenyl)-1-{3-[3-(4-hydroxy-3-methoxy-phenyl)-acryloyl]-phenyl}-propenone was formed at a yield of 35%. JC3 dimer was identified through infrared, high-resolution mass spectroscopy, and nuclear magnetic resonance spectroscopy (NMR) (Figure 6) [21].

### 3.2. Animals

Male Sprague–Dawley rats (6 weeks, 220–220 g) from Samtaco CO. (Osan, Korea) were used for the C/K rat model. Animals were given 1 week to acclimate prior to the experiment and were kept in a under a 12 h light/dark cycle, in an air-conditioned room at 23 ± 5 °C and 55 ± 10% RH. The rats had ad libitum access to water and food. The procedures for the experiments were performed according to the guidelines of NIH animal care and the study protocol was approved by the Animal Care and Use Committee of Ewha Womans University, School of Medicine.

### 3.3. Experimental Arthritis Groups

In the animal arthritis model, the rats were separated in five groups (*n* = 5): the untreated group (NOR), the C/K-induced arthritis control group (ART), the ART group treated with 1 mg/kg of JC3 dimer (ART + JC3dimer_1), the ART group treated with 5 mg/kg of JC3 dimer (ART + JC3dimer_5), and the ART group treated with 10 mg/kg of JC3 dimer (ART + JC3dimer_10). JC3 dimer was given to the mice intraperitoneally (I.P.).

### 3.4. Induction of Arthritis

The C/K-induced arthritic rat model was conducted as previously described [16]. Arthritis was induced using 100 μL of 5% carrageenan/kaolin dissolved in a pyrogen-free sterile saline single injection in the left knee joints. Three behavioral parameters such as weight distribution ratio (WDR), knee thickness, and the use of a joint flexion test was measured in order to evaluate the progression of arthritis for 6 days. JC3 dimer was given (1, 5, and 10 mg/kg, I.P.) 24 h after inducing arthritis and every day for 6 days. The behavioral tests were performed blindly (Figure 7).

### 3.5. Knee Thickness Evaluation

With a dial thickness gauge, the knee thickness of the arthritis induced knee (left knee) was measured one day after arthritis induction. Knee thickness score is expressed against knee thickness measured prior to arthritis induction. This was done every day for 6 days.

### 3.6. Weight Distribution Ratio (WDR)

The WDR is a measure of the percent of the downward force being applied by the hind leg as it bears its weight. As previously described [16], an incapacitance meter (Ugo-basile Biological Research Apparatus Co., Comeria-Varese, Italy) was used to measure the WDR. On top of the incapacitance meter rests a test box with a slanted plan containing two mechanotransducers. A rat was placed inside with each hind limb on top of one mechanotransducer, which measures the weight being borne. Average values of 5 s were measured, and four different measures were then calculated. WDR percentage calculation was done with the following formula: (weight borne by damaged limb/total weight borne by both limbs) multiplied by 100. Normal rats would have a WDR of 50 indicating an equal weight that is carried by one hind paw. This was conducted every day for 6 days.

### 3.7. Squeaking Test

The measure of the squeaking scores was conducted by using a modified method of Yu et al. to evaluate pain and knee rigiditiy during flexion and extension of the hind limb [31]. The number of squeaks vocalized during 10 extension and flexion cycles were recorded. A score rating between 0 (no vocalization) and 1 (vocalization) were assigned. Thus, squeaking scores ranged from 0 to 10 for each hind limb. This was done every day for 6 days.

### 3.8. Histological Assessment

For histological staining using hematoxylin–eosin, the knee joint tissues were processed by fixing the tissues in 10% paraformaldehyde, decalcifying, dehydrating, embedding, and sectioning at 6 µm. To investigate the morphological changes and immune cell infiltration, staining using hematoxylin (Merck, Darmstadt, Germany) and 1% eosin (Sigma-Aldrich Co., MO, U.S.A.) was conducted. Then, the slides were left to air dry and mounted with a cover slip. All slides were observed and acquired with the use of a camera on a microscope (100x magnification, BX51; Olympus Ltd., Tokyo, Japan), and analysis was performed with the use of DP2-BSW software (Olympus Ltd., Tokyo, Japan). Quantitative scoring of the knee joints was done by 3 observers blinded to the experiment.

### 3.9. The Culture and Isolation of FLS

As described previously [32], FLS were isolated from RA patients who were subjected to knee joint replacement therapy. FLS passages 3–6 were used in this study, and culture was done on cells using Dulbecco’s modified Eagle medium (low glucose) with 100 U/mL penicillin, 100 μg/mL streptomycin sulfate, and 10% (vol/vol) fetal bovine serum. Reagents for cell culture were provided by Gibco-Invitrogen (Carlsbad, CA, USA).

### 3.10. Enzyme-Linked Immune-Sorbent Assay (ELISA)

For the in vivo study, rat whole blood samples were obtained and were clotted and centrifuged 20 min at 6500 rpm and stored at −70 °C until use. ELISA kits from BD Biosciences Pharmingen (IL-6 and TNF-α, San Diego, CA, USA) and R&D Systems (PGE_2_, Minneapolis, MN, USA) were used to detect the serum levels of the inflammatory mediators according to the manufacturers’ protocol [33].

For the in vitro study, FLS were stimulated with IL-1β (10 ng/mL; ProSpec, Rehovot, Israel) and cultured with different concentrations of JC3 dimer (1, 5, and 10 μg/mL) for 24 h. Then, culture cell media was collected and centrifuged, and the supernatant was used to analyze for IL-8, IL-6, and PGE_2_ using the above-mentioned commercial ELISA kits. MAPK expression was analyzed in FLS cells after removing the cell culture media and following the protocol provided by the manufacturer for use in cells (MAPK, RayBiotech Inc., Atlanta, GA, USA) [33,34].

### 3.11. Reverse Transcription-Polymerase Chain Reaction (RT-PCR)

With the application of reverse transcription-polymerase chain reaction (RT-PCR), the mRNA expression levels of IL-6 and IL-8 were determined. Total RNA was extracted from FLS cells using TRIzol reagent (Invitrogen Co., Carlsbad, CA, USA). Using reverse transcriptase, complementary DNA was then synthesized from total RNA (Takara Co., Shiga, Japan). Primers used in the study were configured and were produced using mRNA sequences that have been published and Primer3 software (Whitehead Institute for Biomedical Research, Cambridge, MA, USA; www.genome.wi.mit.edu). RT-PCR was performed with a PTC-100 thermal cycler (MJ Research, Inc., Watertown, MA, USA). The PCR products were then separated on 1.2% agarose gels and were stained using ethidium bromide. i-Max™ (CoreBio System Co., Seoul, Korea) was used to analyze the band density, and the expression levels were calculated by determining the density of the bands normalized to glyceraldehyde-3-phosphate dehydrogenase (GAPDH). In Table 1, annealing temperatures and primer sequences have been listed.

### 3.12. Statistical Analysis

One-way ANOVA on SPSS Ver. 13.0 (SPSS; Chicago, IL, USA) was used for statistical analysis, and results were further analyzed with Tukey’s post hoc test to find statistical differences between groups. Results are expressed as means ± SEMs. *p* Values of <0.05 were considered to be statistically significant.

## 4. Conclusions

In conclusion, via inhibition of the p38/ERK MAPK pathway, JC3 dimer can reduce the inflammatory response resulting in anti-arthritic effects. However, one of the limitations of this study is that this was confirmed in C/K-induced arthritis rats while using only behavioral parameters such as pain, edema, and the measurement of inflammatory cytokines. The effects of JC3 dimer on the cell and molecular mechanisms by which JC3 dimer inhibits arthritis remain to be explained in vivo and could be a basis for investigation in future studies. Our results suggest that JC3 dimer has great therapeutic drug resource potential for arthritis treatment. Therefore, further studies should focus on the development of drugs based on JC3 dimers with better efficacy and less side effects in inflammatory and arthritic diseases. 

## Figures and Tables

**Figure 1 molecules-25-03319-f001:**
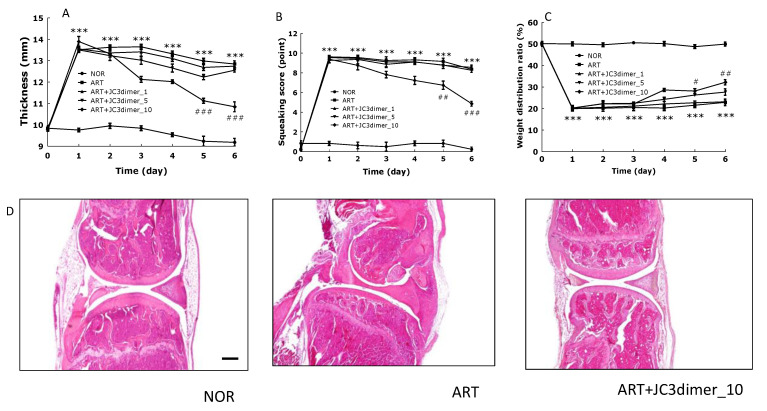
Analysis of arthritic symptoms in the carrageenan/kaolin-induced arthritis rats. Arthritic symptoms were induced by an intra-articular injection of carrageenan/kaolin (100 μL of 5% solution) into the left knees. JC3 dimer was administered orally 1 h before carrageenan/kaolin injection. Arthritic symptoms were determined by (**A**) thicknesses of left knee, (**B**) squeaking scores, (**C**) weight distribution ratios, and (**D**) histological staining (hematoxylin and eosin). NOR: non-treated group; ART: carrageenan/kaolin-treated control group; ART + JC3 dimer_1: carrageenan/kaolin-induced and 1 mg/kg JC3 dimer-treated groups; ART + JC3 dimer_5: carrageenan/kaolin-induced and 5 mg/kg JC3-treated group; and ART + JC3 dimer_10: carrageenan/kaolin-induced and 10 mg/kg JC3-treated group. Results are presented as means ± standard errors. Data analysis was performed using one-way ANOVA followed by Tukey’s post hoc test. *** *p* < 0.001 vs. the NOR group; # *p* < 0.05, ## *p* < 0.01, and ### *p* < 0.001 vs. the ART group.

**Figure 2 molecules-25-03319-f002:**
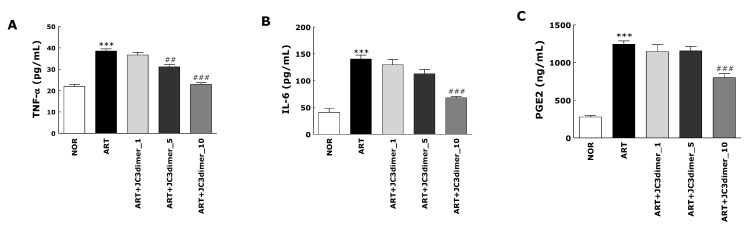
Effects of JC3 dimer on the ELISA determined levels of the pro-inflammatory mediator tumor necrosis factor (TNF)-α (**A**), interleukin (IL)-6 (**B**), and prostaglandin E_2_ (PGE_2_) (**C**) in serum of the rat model of carrageenan/kaolin-induced arthritis. Results are presented as means ± standard errors. *** *p* < 0.001 vs. the NOR group and ## *p* < 0.01 and ### *p* < 0.001 vs. the ART group.

**Figure 3 molecules-25-03319-f003:**
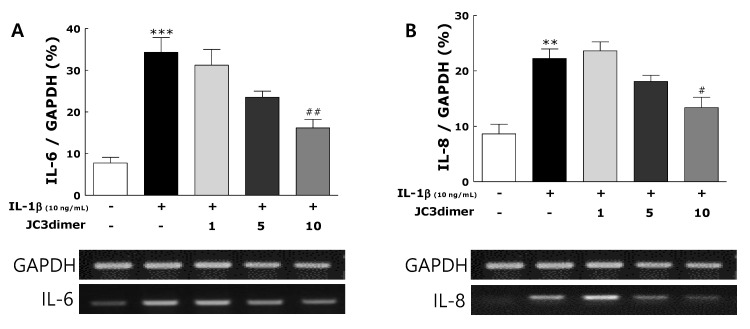
Effects of JC3 dimer on the RT-PCR determined the levels of the pro-inflammatory mediators IL-6 (**A**) and IL-8 (**B**) in cells of the IL-1β-treated fibroblast-like synoviocytes (FLS) cell. NOR: non-treated group; CON: IL-1β-treated control group; IL-1β-treated and 1 μg/mL JC3 dimer-treated groups; IL-1β-treated and 5 μg/mL JC3 dimer-treated group; and IL-1β-treated and 10 μg/mL JC3 dimer-treated group. Results are presented as means ± standard errors. ** *p* < 0.01 and *** *p* < 0.001 vs. the NOR group and # *p* < 0.05 and ## *p* < 0.01 vs. the CON group.

**Figure 4 molecules-25-03319-f004:**
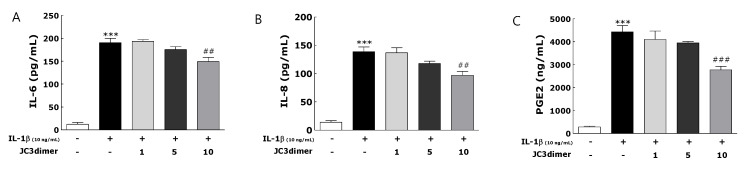
Effects of JC3 dimer (1, 5, 10 μg/mL on the ELISA determined the levels of the pro-inflammatory mediators IL-6 (**A**), IL-8 (**B**), and PGE2 (**C**) in cells of the IL-1β-treated FLS cell. Results are presented as means ± standard errors. *** *p* < 0.001 vs. the NOR group and ## *p* < 0.01 ### *p* < 0.001vs. the CON group.

**Figure 5 molecules-25-03319-f005:**
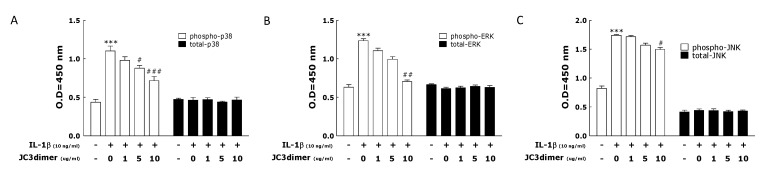
Effect of JC3 dimer on the phosphorylation of p38 (**A**), extracellular signal-regulated kinase (ERK) (**B**) and c-Jun N-terminal kinase (JNK) (**C**) MAPKs in FLS cells previously stimulated by treatment with IL-1β. In the experiments, FLS cells were used for assay 24 hrs after vehicle (medium, NOR) and IL-1β treatments, respectively. JC3 dimer (1, 5, 10 μg/mL was added 30 min before IL-1β treatment, and the cells were used 24 h after IL-1β treatment. MAPKs levels were measured from cells using ELISA. Results are presented as means ± standard errors. Data analysis was performed using one-way ANOVA followed by Tukey’s post hoc test. *** *p* < 0.001 vs. the NOR group and # *p* < 0.05, ## *p* < 0.01, and ### *p* < 0.001 vs. the CON group.

**Figure 6 molecules-25-03319-f006:**
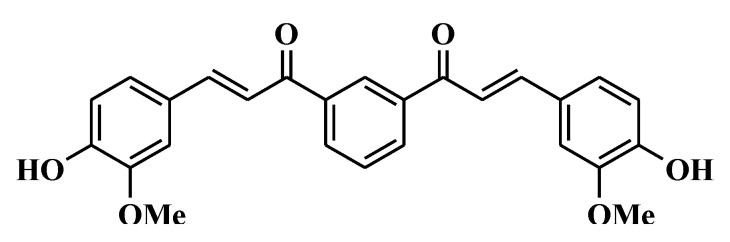
Structure of [(2*E*)-3-(4-hydroxy-3-methoxyphenyl)phenylpro-2-en-l-one] (JC3 dimer).

**Figure 7 molecules-25-03319-f007:**
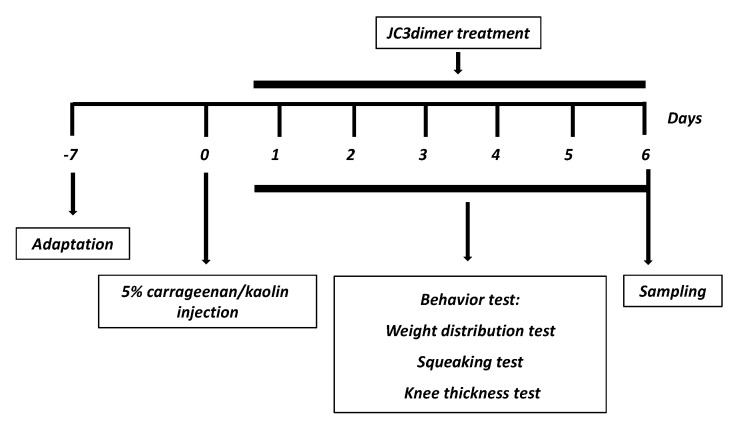
Schematic diagram of the carrageenan/kaolin-induced arthritis experimental schedules.

**Table 1 molecules-25-03319-t001:** Nucleotide sequences of primers and the operating condition of RT-PCR. GAPDH: glyceraldehyde-3-phosphate dehydrogenase.

Primer		Nucleotide Sequence	Condition
GAPDH (201 bp)	sense	5′-acccagaagactgtggatgg-3′	60 °C 30 s, 29 cycles
	antisense	5′-ttctagacggcaggtcaggt-3′	
IL-6 (167 bp)	sense	5′-cagacagccactcacctctt-3′	58 °C 30 s, 28 cycles
	antisense	5′-ctttttcagccatctttgga-3′	
IL-8	sense	5′-gttttgccaaggagtgctaa-3′	58 °C 30 s, 28 cycles
	antisense	5′-ccagacagagctctcttcca-3′	

T: thymine, A: adenine, C: cytosine, G: guanine.

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
