# Peer review of "Benzylideneacetophenone Derivative Alleviates Arthritic Symptoms via Modulation of the MAPK Signaling Pathway"

_molecules, 2020, doi:10.3390/molecules25153319_

Round 1
Reviewer 1 Report
In this paper, the effects of the benzylideneacetophenone derivative 3-(4-hydroxy-3-methoxy-phenyl)-1-{3-[3-(4-13 hydroxy-3-methoxy-phenyl)-acryloyl]-phenyl}-propenone (JC3 dimer) were investigated both in vivo and in vitro using the carrageenan/kaolin (C/K)-induced knee arthritis rat model and fibroblast-like synoviocytes (FLS) from RA patient. In the C/K rat model, JC3 dimer administered at the doses of 1, 5, or 10 mg/kg/day for 6 days, reduced arthritic symptoms of inflammation in the knee joints. In FLS stimulated with IL-1β, JC3 dimer suppressed the increase of IL-8, IL-6, and PGE2. The Authors concluded that JC3 dimer has suppressive effects on arthritis, and that JC3 dimer might be developed as potential agent for arthritis treatments.
I think the manuscript has merits, but the authors should address some questions before publication.
- In the “Introduction” section, the Authors should furnish some information about FLS role in RA
- Has the animal protocol been authorized?
- What was the age and weight of the animals?
- In the “Experimental arthritis groups” section, the route of administration should be reported.
- Data from animal studies, demonstrated that only the dose of 10 mg/kg was effective. Why the Authors did not choose higher doses than 10 mg/kg?
- To confirm the anti-inflammatory effect of JC3 dimer, the level of an anti-inflammatory cytokine such as IL-10 by RT-PCR and ELISA could be investigated in FLS from RA patients.
- To further demonstrate the effective of anti-inflammatory activity of JC3, the expression of IKK and pIKK proteins could be evaluated by western blot analysis in FLS from RA patients.
Author Response
For reviewer 1’s comments:
1. In the “Introduction” section, the Authors should furnish some information about FLS role in RA
Author response: More information about FLS and its role in RA have been provided in the introduction section as has been advised. Please refer to the tracked changes on lines 44-62, 64, 72-74, and 76.
2. Has the animal protocol been authorized?
Author response: This study protocol was approved by the Animal Care and Use Committee of Ewha Womans University, School of Medicine. Therefore, we changed the expression in Animal section as: The procedures for the experiments were performed according to the guidelines of NIH animal care and the study protocol was approved by the Animal Care and Use Committee of Ewha Womans University, School of Medicine.
3. What was the age and weight of the animals?
Author response: The age of the SD rats was 6 weeks and they weighed between 200 to 220 g. This has been edited into line 248.
4. In the “Experimental arthritis groups” section, the route of administration should be reported.
Author response: The route of administration of JC3 dimer was mentioned in 3.4. induction of arthritis as i.p. but we have also added this information in 3.3. experimental arthritis groups as advised in line 259.
5. Data from animal studies, demonstrated that only the dose of 10 mg/kg was effective. Why the Authors did not choose higher doses than 10 mg/kg?
Author response: Thank you for your proposition. We supposed so, but 1, 5, and 10 mg/kg were chosen as we presumed that these doses would be enough to provide a significant effect. Contrary to our expectation, 1 and 5 mg/kg doses have shown no significant effect. However, we considered that results of the concentration 1 and 5 could still be meaningful as a low dose data.
6. To confirm the anti-inflammatory effect of JC3 dimer, the level of an anti-inflammatory cytokine such as IL-10 by RT-PCR and ELISA could be investigated in FLS from RA patients. To further demonstrate the effective of anti-inflammatory activity of JC3, the expression of IKK and pIKK proteins could be evaluated by western blot analysis in FLS from RA patients.
Author response (6&7): We agree that if we analyze IL-10 which is an anti-inflammatory cytokine, IKK, and pIKK proteins, it could contribute to supporting the anti-inflammatory effect of JC3 dimer. However, we think that the current experimental design is sufficient to confirm its anti-inflammatory activity as we have provided both pro-inflammatory mediator measurements in rat serum (TNF-α, IL-6, and PGE2) using ELISA in Fig.2 and FLS cell (IL-6, IL-8, PGE2, MAPKs) using RT-PCR and ELISA in Fig. 3, 4, and 5. PGE2 contributes not only to pain and edema formation but also plays a role in angiogenesis triggered by inflammation (1). IL-6 induces pannus formation which contributes to inflammatory cell infiltration and IL-8 is an angiogenic promoter in RA (2, 3). Angiogenesis then contributes to maintaining chronic inflammation in RA by providing more oxygen to the hyperplastic environment and facilitating the increase of infiltrating cells (4). The MAPK pathway is also a frequently studied pathway in arthritis which is the reason why we focused on this pathway. There are also other studies involving IL-1β stimulation and FLS cells without anti-inflammatory cytokine measurement and also evaluating only the MAPK pathway (5). Not only that, MAPK inhibitors are also being studied as a therapeutic for arthritis (6, 7). This is why we think our experiments for JC3 dimer are adequate.
Reviewer 2 Report
In this paper, the authors investigated the effects of JC3 dimer on the carrageenan/kaolin (C/K)-induced rat arthritis model (in vivo experiment) as well as RA patient-derived FLS model (in vitro experiment). They concluded that JC3 dimer could reduce the inflammatory response via inhibition of the p38/ERK MAPK pathway.
However, due to serious concerns about the experimental procedure of the most crucial experiment in this conclusion, which is phosphorylation experiments of MAPKs, their claim is entirely unacceptable.
Specifically,
Comment 1:
Lines151-157: Figure 5 legend
The authors investigated phospho-MAPKs levels in FLS cells after stimulation with IL-1b for 24 hours. This experimental protocol is entirely wrong. Phosphorylation of MAPKs is a phenomenon that is usually observed transiently up to about 10-30 minutes after inflammatory stimulation and returns to almost unstimulated control levels 1 hour after stimulation. However, they measured phosphorylated MAPKs using samples 24 hours after IL-1b stimulation, which is inappropriate for detecting MAPK activation.
Comment 2:
Lines280-283: Culture cell media was then collected, centrifuged, and the supernatant was used to analyze for IL-8, IL-6, PGE2, and MAPK expression by the above mentioned commercial ELISA kits (MAPK, RayBiotech Inc., GA, USA)
No MAPKs are present in the cell culture supernatant.
Author Response
1. Lines151-157: Figure 5 legend. The authors investigated phospho-MAPKs levels in FLS cells after stimulation with IL-1β for 24 hours. This experimental protocol is entirely wrong. Phosphorylation of MAPKs is a phenomenon that is usually observed transiently up to about 10-30 minutes after inflammatory stimulation and returns to almost unstimulated control levels 1 hour after stimulation. However, they measured phosphorylated MAPKs using samples 24 hours after IL-1b stimulation, which is inappropriate for detecting MAPK activation.
Author response: This can be a potential limitation to our study. However, as stated in the book Contemporary Targeted Therapies in Rheumatology by Smolen and Lipsky: “Despite the decrease over time, the amount of phosphorylated MAPKs remains slightly above normal for up to 24 hours after stimulation” (8). Not only that, various studies have also used 24 hours to stimulate cells and detect MAPKs. Such studies include IL-1β-induced inflammation in chondrocytes (9), ultraviolet B-induced human dermal fibroblasts (10), lipopolysaccharide and human bone marrow mesenchymal stem cells (HBMSC) (11), and in induced apoptosis in FLS cells (12), to name a few. Thus, we think that using 24 hr is acceptable for the study.
2. Lines280-283: Culture cell media was then collected, centrifuged, and the supernatant was used to analyze for IL-8, IL-6, PGE2, and MAPK expression by the above mentioned commercial ELISA kits (MAPK, RayBiotech Inc., GA, USA). No MAPKs are present in the cell culture supernatant.
Author response: We apologize for the confusion that was caused by our methodology writing. The supernatant was used to analyze for IL-8, IL-6, and PGE2. MAPK expression was analyzed on FLS cells after removing the cell culture media and following the protocol provided by the manufacturer for use in cells. We have edited the journal accordingly on lines 292-294.
Reviewer 3 Report
The authors describe the anti-inflammatory effect of a novel synthetic compound, JC3, in a rat model of RA. They show the amelioration of the arthritis in the JC3-treated rats, based on limb thickness, squeeking score and WDR as well as histological analysis. The JC3-treated group showed decreased inflammatory mediator production. Another set of data came from human RA FLS cell experiments. Here, they found decreased IL-6 and IL-8 expression, then decreased production of IL-6,-8 and PGE2. In the background of these anti inflammatory effects the authors suspect the down regulation of the MAPK pathways.
Combining the results from rat model and human FLS cells shows that the treatment could be effective in humans as well increasing the scientific value of the study. The paper is logically written, easy to follow and clearly formatted.
Some questions:
- In the legend of Fig. 1 the authors write: "Arthritic symptoms were induced by an intra-plantar injection of carrageenan/kaolin (100 μl of 5% solution) into the left knees." this is obviously contradicting.
- In 2.4 the authors wrote: "JC3 dimer treatment (10 μg/ml) significantly inhibited the increased MAPK phosphorylation in stimulated FLS." It should be stated that this effect was different to the 3 parallel MAPK pathways. What could explain that the JNK was the least affected?
- The last 3 paragraphs of 2.4 is somewhat confusing. The authors skip from their own results to literature data and this is not clearly indicated. Please make sure that the reader can clearly distinguish results of this study from other publications.
- For the animal experiments do the authors have a specific permission number?
- In 3.3 the authors wrote:"In the animal arthritis model, the rats were separated in four groups (n=5):", in fact, there were 5 groups in all figures.
- In 3.4 please correct "100uL".
- In 3.5 As another negative control the untreated knee should also be measured.
- In 3.7 "The number of squeaks vocalized during five 5-second extension and flexion cycles were recorded. A score rating between 0 (no vocalization) and 1 (vocalization) were assigned. Thus, squeaking scores ranged from 0 to 10 for each hind limb." If the above description is correct then both hind limbs could get a score between 0-5.
- In 3.8 for the histology did they decalcify the tissues?
- In Table 1 I don’t think it is necessary to list the DNA bases.
Overall this is an interesting study carrying scientific value. However, in the discussion the potential effect of the JC3 on other cell types could be at least mentioned and it would be interesting to see some speculation how the neutrophils or macrophages could be affected by JC3. I would doubt that the in vivo anti-inflammatory effect would exclusively be the result of FLS targeting.
Author Response
1. In the legend of Fig. 1 the authors write: "Arthritic symptoms were induced by an intra-plantar injection of carrageenan/kaolin (100 μl of 5% solution) into the left knees." this is obviously contradicting.
Author response: Thank you for pointing this out. Intra-plantar injection was a writing error on our part as we also do carrageenan injections on the footpad for hyperalgesia experiments. We have corrected it to intra-articular on line 113.
2. In 2.4 the authors wrote: "JC3 dimer treatment (10 μg/ml) significantly inhibited the increased MAPK phosphorylation in stimulated FLS." It should be stated that this effect was different to the 3 parallel MAPK pathways. What could explain that the JNK was the least affected?
Author response: Thank you for your suggestion and we have revised the effects of JC3 on the three MAPKs separately on lines 164 and 165. Our study is the first experiment on JC3 dimer in relation to arthritis. We have initially demonstrated that JC3 dimer has potential anti-inflammatory and anti-arthritic effects. However, the MAPK pathway has multiple possible targets that lead to nuclear translocation. We are interested in exploring how JC3 dimer clearly works
3. The last 3 paragraphs of 2.4 is somewhat confusing. The authors skip from their own results to literature data and this is not clearly indicated. Please make sure that the reader can clearly distinguish results of this study from other publications.
Author response: We apologize for the confusion. We have revised the discussion chapter in chapter 2 (results and discussion) to make it distinct. The discussion chapter is on lines 174 to 234.
4. For the animal experiments do the authors have a specific permission number?
Author response: This study protocol was approved by the Animal Care and Use Committee of Ewha Womans University, School of Medicine (#MRI 10-04). Therefore, we changed the expression in Animal section as: The procedures for the experiments were performed according to the guidelines of NIH animal care and the study protocol was approved by the Animal Care and Use Committee of Ewha Womans University, School of Medicine. line 252
5. In 3.3 the authors wrote: "In the animal arthritis model, the rats were separated in four groups (n=5):", in fact, there were 5 groups in all figures.
Author response: The number of groups has been corrected and is reflected on line 255.
6. In 3.4 please correct "100uL".
Author response: The unit for 100 μL has been corrected and is reflected on line 262.
7. In 3.5 As another negative control the untreated knee should also be measured.
Author response: We would like to ask if by negative control, does it mean non-C/K induced arthritis knee without JC3 dimer treatment? In our study, we used the NOR group as the negative control as they were not induced with arthritis and they also did not receive JC3 dimer treatment. We would appreciate it if this can be clarified for us.
8. In 3.7 "The number of squeaks vocalized during five 5-second extension and flexion cycles were recorded. A score rating between 0 (no vocalization) and 1 (vocalization) were assigned. Thus, squeaking scores ranged from 0 to 10 for each hind limb." If the above description is correct then both hind limbs could get a score between 0-5.
Author response: Thank you for pointing out what we have overlooked. We wanted to give a more detailed explanation for this method but we made it more straight to the point in our revision. It should be ten times extension and flexion cycles. This has been reflected in the revised paper on line 287.
9. In 3.8 for the histology did they decalcify the tissues?
Author response: The rat knee joints were decalcified 3 days followed by washing, dehydration, and the succeeding steps for histological processing. We have added this in the revised paper on line 292.
10. In Table 1 I don’t think it is necessary to list the DNA bases.
Author response: Thank you for your feedback. However, we would prefer to keep Table 1 as it can be needed as a reference by other readers. If this is really unnecessary, we can supply this table as a supplementary data.
Round 2
Reviewer 2 Report
I am satisfied with the answers and have no complaints about the paper.